# Chemotherapeutic Drug-Regulated Cytokines Might Influence Therapeutic Efficacy in HCC

**DOI:** 10.3390/ijms222413627

**Published:** 2021-12-20

**Authors:** Chun-I Wang, Pei-Ming Chu, Yi-Li Chen, Yang-Hsiang Lin, Cheng-Yi Chen

**Affiliations:** 1Radiation Biology Research Center, Institute for Radiological Research, Chang Gung Memorial Hospital, Chang Gung University, Taoyuan 333, Taiwan; yeewang0330@gmail.com; 2Department of Anatomy, School of Medicine, China Medical University, Taichung 404, Taiwan; pmchu@mail.cmu.edu.tw; 3Department of Cell Biology and Anatomy, College of Medicine, National Cheng Kung University, Tainan 70101, Taiwan; danielj30707@gmail.com; 4Liver Research Center, Chang Gung Memorial Hospital, Linkou, Taoyuan 333, Taiwan; yhlin0621@cgmh.org.tw

**Keywords:** chemotherapy, drug resistance, cytokine, HCC

## Abstract

Hepatocellular carcinoma (HCC), the most common type of liver cancer, is the second leading cause of cancer-related mortality worldwide. Processes involved in HCC progression and development, including cell transformation, proliferation, metastasis, and angiogenesis, are inflammation-associated carcinogenic processes because most cases of HCC develop from chronic liver damage and inflammation. Inflammation has been demonstrated to be a crucial factor inducing tumor development in various cancers, including HCC. Cytokines play critical roles in inflammation to accelerate tumor invasion and metastasis by mediating the migration of immune cells into damaged tissues in response to proinflammatory stimuli. Currently, surgical resection followed by chemotherapy is the most common curative therapeutic regimen for HCC. However, after chemotherapy, drug resistance is clearly observed, and cytokine secretion is dysregulated. Various chemotherapeutic agents, including cisplatin, etoposide, and 5-fluorouracil, demonstrate even lower efficacy in HCC than in other cancers. Tumor resistance to chemotherapeutic drugs is the key limitation of curative treatment and is responsible for treatment failure and recurrence, thus limiting the ability to treat patients with advanced HCC. Therefore, the capability to counteract drug resistance would be a major clinical advancement. In this review, we provide an overview of links between chemotherapeutic agents and inflammatory cytokine secretion in HCC. These links might provide insight into overcoming inflammatory reactions and cytokine secretion, ultimately counteracting chemotherapeutic resistance.

## 1. Introduction

Hepatocellular carcinoma (HCC), the most common type of primary liver cancer, is the most aggressive malignancy, with a median survival time of 7 to 9 months worldwide [1]. The mortality and incidence of HCC have even increased in the last decade [2,3]. Currently, HCC causes approximately one million deaths annually. Advanced HCC is associated with a high recurrence rate and a short survival time [4]. In fact, approximately 80% of HCC patients have advanced HCC and have a median survival time of less than 1 year from diagnosis [2]. HCC usually develops via the progression of cirrhosis and chronic liver diseases. Several risk factors, such as alcohol consumption, viral infection, and toxin exposure, mediate HCC development [5]. Surgery followed by chemotherapy is the most common curative therapeutic regimen for HCC [6]. However, surgical resection of HCC has several difficulties and limitations due to the distribution and size of tumors in the liver and surrounding blood vessels. Moreover, more than two-thirds of HCC patients have advanced-stage disease with metastasized tumor cells, and it is highly difficult to completely remove these tumor cells surgically [3,7].

Transarterial chemoembolization (TACE) is a frequently utilized treatment of several locoregional therapies proposed for nontransplantable and unresectable HCC, and the response rates range from 10% to 50% [8]. Previous studies have demonstrated that HCC patients with TACE often result in some inflammatory cytokines secretion, which occurs as a result of hepatic tissue injury [9]. Several inflammatory cytokines, such as IL-5, IL-6, and IL-17A, were higher in the serum of HCC patients than in healthy controls. However, IL-22 and IL-1b levels were lower in HCC patients [10]. Patients with larger tumors (>5 cm) displayed a significant elevation in IL-6 levels at early phase coupled with post-TACE hepatitis, as well as increases in IL-4, IL-5, and IL-10 levels at late phase after TACE [10]. Inflammation has been defined as a critical factor for tumor recurrence [11]. IL-6 receptor alpha (IL-6Rα), a multifunctional cytokine, plays key roles in inflammation and HCC development [12].

Previously, Walter and colleagues reported that a systemic therapeutic approach targeting advanced HCC with low-dose chemotherapeutic agents, such as rofecoxib, pioglitazone, and capecitabine, has been evaluated in patients with incurable HCC. In total, 38 HCC patients were evaluated in this one-arm, multicenter phase II trial [13]. Moreover, advanced HCC usually has a poor prognosis, and systemic therapy with cytotoxic agents has a limited effect [14]. To date, several traditional cytotoxic chemotherapeutic agents, including 5-fluorouracil (5-FU), cisplatin, doxorubicin, paclitaxel, and mitomycin, have been used to treat HCC patients; however, the effects have been limited by systemic toxicity and acquired resistance of the tumor after treatment [15,16,17].

Inflammation has been demonstrated to be closely related to the initiation and development of HCC [4]. Previously, tumor necrosis factor-α (TNF-α), a critical inflammatory mediator, was demonstrated to be a potential therapeutic target in numerous cancers [4]. Moreover, the levels of various inflammatory cytokines, such as TNF-α, interleukin (IL)-1, and IL-6, are significantly higher in the serum of HCC patients than in that of healthy controls [18,19]. A previous study showed that M2 macrophages release all three of the abovementioned cytokines. Tumor-associated macrophages (TAMs) have also been demonstrated to increase tumor size, angiogenesis, intrahepatic metastasis, and the recurrence rate via the STAT3 signaling pathway in HCC cell lines [20,21]. Wang and colleagues reported that a high level of TNF-α is a predictor of poor survival in patients with HCC, as shown by survival and Cox regression analyses. Infliximab, an anti-TNF-α antibody, can increase Fluorouracil-induced levels of cleaved caspase-3 in the presence of an active complement in HCC [4]. Furthermore, blocking TNF-α production could be a suitable approach to enhance the effect of classical chemotherapy in HCC patients, especially those who have a modest response to classical chemotherapy [4]. Thus, the inflammatory cytokine TNF-α could also be used as a biomarker to facilitate the early diagnosis of HCC. Chronic inflammation plays a crucial role in cancer initiation and progression [22] and a well-established role in the development of HCC, often in association with liver fibrosis and cirrhosis. In addition, the activity of transforming growth factor-beta (TGF-β) has been established as essential for aspects of HCC pathogenesis, including the activation of cancer-associated fibroblasts (CAFs) [23,24,25].

Numerous HCC patients can develop chronic liver injury or inflammation, indicating that HCC is a carcinogenic process based on inflammation [26]. Chemokines can modulate the response of immune cells that migrate to proinflammatory stimuli, to influence inflammation-mediated tumorigenesis [27]. In this review, we provided an overview of the effects of numerous chemotherapeutic drugs on cytokine/chemokine secretion, which can indicate whether the tumor microenvironment plays a critical role in the chemotherapeutic response. We reviewed numerous inflammatory cytokines influenced by various drugs and discussed the effects of chemotherapeutic agents on cytokine secretion and the tumor microenvironment. These observations might provide insight into drugs that could be used alone or in combination with others in treating HCC.

## 2. Fluorouracil (5-FU)

Chemotherapy regimens included 5-fluorouracil (5-FU) with other drugs, but patients exhibited low radiological response [28]. Among different systemic treatments, including 5-fluorouracil and doxorubicin, have been used in a limited number cases such as nontransplantable or nonresectable patients [29]. 5-FU can inhibit cell proliferation by forming fluorodeoxyuridine monophosphate via blocking of thymidylate synthase, which can catalyze the synthesis of the DNA precursor thymidylate (Table 1) [30]. TNF-a has been identified as an independent predictor of poor survival in patients with HCC. Therefore, anti-TNF-a treatment with 5-FU can induce HCC tumor cell apoptosis via antibody-dependent cellular cytotoxicity (ADCC) and complement-dependent cytotoxicity (CDC) processes (Table 1) [4]. The higher chemoresistance to doxorubicin, 5-FU, and cisplatin is observed in anoikis-resistant (AR) cells than adherent HCC cells. The lower expression of E-cadherin and higher expression of N-cadherin and vimentin were exhibited in AR HCC cells compared with adherent HCC cells (Table 1) [31]. Additionally, 17β-Estradiol (E2) decreased the IL-6/STAT3 signaling to attenuate the AR HCC cell proliferation (Table 1) [31].

## 3. Cisplatin

Cisplatin is an effective and broad-spectrum chemotherapeutic drug for treating HCC. However, several side effects of cisplatin are displayed. In addition, long-term treatment with cisplatin can cause chemoresistance, which attenuates the clinical application of cisplatin with a limited range [32]. Previously, cisplatin has been demonstrated to activate the ATM-NF-kB pathway in a TonEBP-dependent manner (Table 1). Additionally, several proinflammatory cytokines mediated by cisplatin were blocked after silencing of TonEBP or XPF expression. Cisplatin has been displayed to increase DNA crosslink formation to induce inflammation by the ATM-NF-kB signaling pathway through the TonEBP-ERCC1/XPF complex (Table 1). Cisplatin also induces the interaction between chromatin and the ERCC1/XPF dimer in a TonEBP-dependent manner, resulting in DNA repair and cisplatin resistance [33]. YC-1, an anti-cancer drug, induced cancer cell death that can be reversed by overexpression of STAT3. Moreover, YC-1 can decrease STAT3 activity by increasing the cisplatin-induced polyubiquitination of p-STAT3(705). In summary, YC-1 has been determined to play a novel anticancer role to enhance the HCC cell chemosensitivity to cisplatin in a STAT3-dependent manner (Table 1) [34]. Furthermore, STAT3 has been determined to be associated with drug resistance [35]; tumor growth was reduced and cisplatin-induced chemo-cytotoxicity was enhanced after silencing of STAT3 expression. Therefore, the STAT3 pathway may be a potentially effective anticancer target (Table 1) [36]. Cisplatin also induced CKLF1 expression to create an aggravating inflammatory environment, which facilitates tumor growth and cisplatin-resistance [32]. Chemokines can affect the HCC occurrence and development in various ways, including inflammation and the impact on immune cells [37]. CKLF1 is newly identified chemokine and plays a key role in various diseases [38]. Minocycline, a semisynthetic tetracycline and a highly lipophilic molecule, causes cell cycle arrested at S phase and increased apoptotic rate associated with numerous molecule dysregulation, including p27, cleaved-PRAP-1, cleaved-caspase8, and cleaved-caspase 3(Table 1) [39]. Cisplatin modulates various pathways, such as the ATR, p53, p73, and MAPK signaling, to elicit a sequential responses in the cell, including DNA repair, drug resistance, and apoptosis (Table 1) [40].

## 4. Oxaliplatin

Oxaliplatin, a platinum chemotherapeutic drug with relatively few side effects, has been extensively used to reduce tumor recurrence and increase the survival rate in HCC [41]. However, chemoresistance to oxaliplatin is observed to decrease HCC cell apoptosis [42]. Wu and colleagues demonstrated that IL-17/IL-17 receptor (IL-17R) levels in both patients with HCC and HCC cell lines are increased by oxaliplatin treatment. IL-17/IL-17R binding inhibited oxaliplatin-induced apoptosis and induced autophagy in HCC cell lines. Moreover, the levels of autophagy-related molecules were increased by IL-17/IL-17R binding, and autophagy was shown to induce oxaliplatin resistance in HCC patients [43] (Table 2).

Rather, IL-17 signals through nuclear factor (NF)-κB [44], mitogen-activated protein kinase (MAPK) and phosphoinositide 3-kinase (PI3K) [45] signaling pathways. Wu and colleagues have demonstrated that the increased Bcl-2 and decreased Bax are observed in HCC cells using Western blot after oxaliplatin treatment with IL-17 stimulation. However, the effect is abolished after stimulation with anti-IL-17 antibody. Moreover, IL-17 can induce the levels of p-JAK2 and p-STAT3 with oxaliplatin treatment. Based on the evidence, IL-17 interacts with IL-17R and can decrease oxaliplatin-induced cell death through the JAK2-STAT3 cascade [43]. There are several pathways influenced by IL-17 signals, such as Janus kinase 2 (JAK2)/STAT3, that possess a crucial role in regulating a number of processes related to tumorigenesis, including cell cycle progression, apoptosis, and tumor cell metastasis (Table 1) [46]. This evidence implies that IL-17/IL-17R-induced resistance to oxaliplatin in patients with HCC may be acquired through the regulation of autophagy. Moreover, these findings may help to develop approaches to counteract chemoresistance in HCC.

IL-17, a T helper 17 (Th17) cell-secreted cytokine, has been shown to be involved in the pathogenesis and progression of inflammatory diseases [41]. IL-17R is expressed on the surface of various cells, including fibroblasts, epithelial cells, macrophages, and T lymphocytes [47,48]. Moreover, studies in patients with persistently higher levels of IL-17 have consistently indicated that these patients need to receive longer courses of chemotherapy due to their higher recurrence rate [49]. IL-17 can also interact with IL-17R to influence autoimmune and inflammatory diseases, such as rheumatoid arthritis, psoriasis, and systemic lupus erythematosus [50]. Oxaliplatin produces high levels of reactive oxygen species (ROS) in HCC cells, and Oxaliplatin might induce cell apoptosis via the p53-caspase 8-caspase 3 cascade [51]. Oxaliplatin is a third-generation platinum-based chemotherapeutic drug that possesses the antitumor activity [52]. In addition, oxaliplatin also induces the inflammatory activity and the secretion IL-6 cytokine in HCC cells via nuclear factor kappa B (NF-κB) and p38 mitogen-activated protein kinase (MAPK) signaling pathways (Table 1) [53].

## 5. Celecoxib

Previously, elevated levels of IL-6 and the IL-6 receptor (IL-6R) have been reported to be highly correlated with STAT3 activation in HCC cell lines. Liu et al. showed that celecoxib induces HCC cell apoptosis and inhibits STAT3 phosphorylation by reducing Janus activated kinase (JAK2) phosphorylation. IL-6-induced phosphorylation and nuclear translocation of STAT3 are also blocked by celecoxib. Furthermore, HCC cell viability was found to be reduced more significantly when celecoxib treatment was combined with sorafenib or doxorubicin [54] (Table 2).

Treatment with celecoxib possesses the ability to promote apoptosis, inhibit cell proliferation, and induce cell cycle arrest in HCC cells through the upregulation of E-cadherin protein via the inhibition of the Cyclooxygenase-2 (COX-2) prostaglandin E2 (PGE2)-p-Akt/p-ERK cascade (Table 1) [55]. COX-2, a rate-limiting enzyme, plays important roles in the process of inflammation-tumor transformation and the sequential oxygenation of arachidonic acid (AA) to synthesize prostaglandins and thromboxanes [56]. Celecoxib is a selective COX-2 inhibitor that moderates portal hypertension and liver fibrosis by suppressing gut-liver inflammation [57] and epithelial–mesenchymal transition of hepatocytes (Table 1) [58]. COX-2, an upregulated cancer-related inflammatory mediator in numerous tumors, is defined as a prognosis indicator in many cancer types [59]. Additionally, the expression of COX-2 in tumor tissue is significantly correlated with various inflammatory cells (Table 1) [60]. Therefore, we conclude that celecoxib might serve as a therapeutic agent for HCC by suppressing the IL-6/STAT3 signaling pathway and could be combined with other chemotherapeutic drugs to overcome drug resistance.

## 6. Doxorubicin

To date, doxorubicin, a chemotherapeutic drug for advanced HCC, has exhibited low efficacy, with a response rate of 15–20% [61]. However, numerous lines of evidence have shown that doxorubicin plays a role in accelerating malignant cancer cell progression. For example, doxorubicin induces extracellular matrix degradation, epithelial–mesenchymal transition, and tumor invasion via the regulation of MMP-2 and MMP-9 enzyme activity and TGFβ signaling activation [62,63] (Table 2). Recently, accumulating evidence has shown that several molecules, such as the inflammatory cytokine IL-6 and the transcription factors NF-kB and STAT3, play important roles in HCC development [2,64,65] (Table 2). Moreover, the IL-6 level is significantly higher in patients with stage III HCC than in patients with HCC of other stages [64]. STAT3, a crucial signal transducer that modulates IL-6 signaling in the nucleus, is highly correlated with the prognosis of HCC patients [66]. This evidence indicates that the IL-6/STAT3 signaling cascade may be a therapeutic target in controlling HCC progression. Liu et al. reported that CKLF1 might accelerate the development and progression, as well as the metastasis and proliferation, of HCC by dysregulating the IL-1/STAT3 cascade. Additionally, CKLF1 can induce doxorubicin resistance in HCC cells by inhibiting apoptosis via IL-6/STAT3 signaling [67].

The long noncoding RNA (lncRNA) *H19* represents tumor-promoting or tumor-suppressive actions and is regulated under inflammatory conditions (Table 1) [68]. Moreover, *H19* can attenuate cell survival and proliferation after doxorubicin stimulation using clonogenicity and proliferation assays, suggesting *H19* possesses chemosensitizing actions [68]. HCC cells with higher expression levels of cytochrome p450-3A4 (CYP3A4) enzyme exhibited doxorubicin chemoresistance in a cirrhosis-dependent manner. CYP3A4 plays roles in reducing systemic doxorubicin toxicity and inducing cell death in HCC cells with combined treatment of doxorubicin and sorafenib (Table 1) [69]. Therefore, CYP3A4 expression might potentially be defined as an indicator to predict chemotherapeutic response [70]. In-depth research displayed that syncytin-1-promotes hepatocarcinogenesis may through the inflammation-activated MEK/ERK pathway. Syncytin-1 also blocks MEK/ERK pathway to suppress cell apoptosis induced by doxorubicin [71]. Syncytin-1 is overexpressed in various types of cancers, including leukemia, endometrial cancer, and breast cancer [72,73]. In the process of HCC development, numerous inflammation-associated pathways are activated, such as the mitogen-activated protein kinase (MEK) and extracellular signal-regulated protein kinase (ERK) pathways [74,75]. Based on these results, the relationship between HCC development and inflammation-mediated carcinogenesis is illustrated, and potential biomarkers involved in the inflammation-related pathway modulated by doxorubicin treatment might be therapeutic targets for HCC.

## 7. Sunitinib

Zhu et al. found that higher levels of inflammatory molecules, such as IL-6, were associated with a poor outcome. Sunitinib shows evidence of antitumor activity in advanced HCC, with modest adverse effects. Rapid changes in circulating inflammatory cytokines are potential modulators of the response and resistance to sunitinib in HCC [76]. Emerging data indicate that inflammatory signaling pathways and/or immune cells induce tumor angiogenesis [77,78,79]. Inflammation induced by numerous etiologies, such as hepatitis [80], is another key feature of HCC [81]. Sunitinib is an oral multitargeted receptor tyrosine kinase inhibitor (TKI) that is also approved for the treatment of imatinib-resistant gastrointestinal stromal tumors and renal cell carcinomas [82,83]. Sunitinib has been demonstrated to inhibit various molecules, including VEGFR1-3, PDGFRα, PDGFRβ, stem cell factor receptor (KIT), and FMS-like tyrosine kinase 3 [84]. Additionally, these pathways have been reported to be involved in inflammation and angiogenesis. Sunitinib, which delays tumor progression, is highly correlated with decreased circulating levels of the inflammatory molecule IL-6 and soluble c-KIT. Moreover, Zhu et al. reported that higher levels of IL-6 and soluble SDF1α are associated with rapid progression or mortality with sunitinib treatment in patients with advanced HCC [76] (Table 2). Hence, inflammatory-associated factors, such as IL-6 and SDF1α, might not only play roles in tumor progression on this therapy but also be potential novel targets for HCC. Inflammation has been reported to play an important role in tumor initiation and progression [85]. Moreover, dysregulation of proinflammatory cytokines in the tumor microenvironment has been demonstrated to influence metastasis by inhibiting proapoptotic host immune defense mechanisms and repressing metastasis suppressors [86]. In conclusion, the inflammatory tumor microenvironment, gradually formed by tissue hypoxia and induced by the generation of numerous inflammatory cells and cytokines, has a promotive role in HCC metastasis. Therefore, the control of inflammation might be important for improving treatment outcomes in advanced HCC.

## 8. Sorafenib

Sorafenib, a multitargeted TKI, is the first agent to demonstrate a significant improvement in the median overall survival time in patients with advanced HCC and has been approved by the US Food and Drug Administration (FDA) for systemic therapy [87,88]. Sorafenib may exert its antivascular effects by targeting VEGFR2, VEGFR3, and PDGFβ receptors and may block tumor cell proliferation by targeting the RAF/MEK/ERK cascade [89,90]. Recently, three new multikinase inhibitors—lenvatinib [91], regorafenib [92], and cabozantinib [93]—have been approved by the FDA for advanced HCC [94], but the median overall survival time of patients treated with these drugs needs to be assessed.

The chemokines CCL22 and CCL17 are upregulated by sorafenib in HCC. Mechanistically, sorafenib induces CCL22 expression through the TNF-α-RIP1-NF-κB cascade (Table 1) [95]. The macrophage-derived CCL22 and thymus-regulated CCL17, have been found to interact with their receptor CCR4 to influence cell migration [96]. Activated (M2) macrophages might be a critical factor to contribute to poor prognosis in HCC and induces tumor cell invasion through epithelial-to-mesenchymal transition (EMT) induced by CCL22, which implies that CCL22 is highly expressed both in tumor and stromal cells [97]. Thus, CCL22 might be a target for clinical application to alleviate sorafenib resistance. The levels of IL-6Rα are induced after sorafenib treatment. Moreover, IL-6-induced tumor growth of HCC cells via STAT3 phosphorylation at tyrosine 705 in the presence of sorafenib is reversed with IL-6Rα depletion (Table 1) [12].

The Octamer-binding transcription factor 4 (OCT4) is modulated by inflammatory cytokine interleukin-6 (IL-6) and is highly associated with tumor recurrence and poor prognosis of HCC (Table 1). The expression of DNA methyltransferase (DNMT) has been determined to be highly associated with OCT4 expression and drug resistance in HCC, and the expression levels of OCT4 are positively correlated with the IL-6 levels in serum. Furthermore, the panel of OCT4, DNMT3b, and IL-6 can be defined as markers to predict HCC recurrence and poor prognosis [98]. Emerging evidence has shown a correlation between OCT4 expression and tumor initiation factors and cancer stem cell-like phenotypes in numerous cancers, including HCC, prostate cancer, and melanoma [99]. The DNMT1 expression regulated by OCT4 was further analyzed using OCT4 overexpression and DNMT1 silencing. Moreover, the expression of OCT4 is decreased with DNMT silencing in sorafenib-resistant HCC cells in the presence of IL-6 or not [98]. Taken together, we suggest that DNMT possesses a vital role in the OCT4 expression mediated by IL-6 and the drug sensitivity of sorafenib-stimulated HCC. The activation level of STAT3 modulates DNMT/OCT4, which confers tumor recurrence and prognosis in patients with HCC [98].

## 9. Infliximab

Infliximab is an FDA-approved anti-TNF-α monoclonal antibody that neutralizes the biological activity of TNF-α by abolishing its effective binding to its receptors [100,101] (Table 2). TNF-α plays important roles not only in cytotoxic effects but also in cytokine network regulation. Accumulating evidence shows that treatment with infliximab+5-FU can prolong overall survival by blocking TNF-α secretion [4]. Based on this evidence, the level of secreted TNF-α plays a crucial role in influencing the therapeutic efficiency of infliximab or infliximab+5-FU. 5-FU is usually administered as a first-line treatment for advanced HCC, but its efficiency needs to be improved by overcoming limitations caused by drug resistance [102]. Hence, combination therapy with 5-FU and other drugs is a possible approach to improve the therapeutic efficiency of 5-FU. Emerging evidence has shown that infliximab treatment synergizes with 5-FU treatment to increase HCC cell apoptosis both in vitro and in vivo, suggesting that HCC development is highly related to the accumulation of inflammatory cytokines [103,104].

Currently, TNF-α level has been demonstrated to be implicated in HCC progression, as displayed by the significantly prolonged survival curve in a mouse model in vivo. Moreover, several pro-inflammatory cytokines, including TNF-α, IL-1β, IL-6 and IL-17, and induced cell apoptosis are decreased with anti-TNF-α treatment in HCC tumor cells (Table 1) [105]. Anti-TNF-α treatments can facilitate cell death and reduce the expression levels of pro-inflammatory cytokines to attenuate HCC tumor progression [105]. Infliximab, a TNF-α inhibitor, is an anti-TNF-α monoclonal antibody, which possesses its ability to influence cell lysis in tumors [106]. However, several pro-inflammatory cytokines, including IL-1β, IL-6, and IL-17, are blocked by infliximab stimulation or combined treatment of infliximab and TNF-α, which indicates that anti-TNF-α treatment might modulate tumor-influenced inflammation in HCC [105]. In summary, infliximab can delay tumor growth and prolong survival time, hence, infliximab might be a suitable chemotherapeutic drug for HCC.

## 10. Galunisertib

Recently, several lines of evidence have confirmed the efficacy of galunisertib, a promising drug under clinical investigation for the treatment of patients with HCC. Furthermore, TGF-β1 reduces the expression of E-cadherin at cell–cell contact sites to increase tumor invasion, but this effect can be reversed by galunisertib [107] (Table 2). TGF-β activity depends strongly on the association with different cytokines and cell types. Moreover, accumulating evidence indicates that TGF-β functions as either a tumor-suppressive regulator or protumorigenic factor in different stages of HCC development [108]. Additionally, the expression of E-cadherin is reduced by TGF-β1 in cell–cell adhesion, which increases cell motility; however, the effects are abolished after being stimulated with galunisertib (Table 1) [109]. Additionally, the mRNA expression of SKI-like (SKIL) and prostate transmembrane protein androgen induced 1 (PMEPA1) is identified to be elevated in HCC tumor tissues compared with controls using a next-generation sequencing approach, and positive correlation with TGF-β1 mRNA concentrations in HCC tissues is observed. However, these genes were strongly abolished by stimulation with galunisertib [110]. The SMAD transcriptional corepressor SKI-novel (SNON), which is encoded by the human SKI-like (*SKIL*) gene, is a TGF-β signaling antagonist. SNON is removed from the response element of *SKIL* gene promoter in the presence of TGF-β signaling, and then the activated SMAD complexes induce *SKIL* gene expression by binding to the promoter [111]. Prostate transmembrane protein androgen induced 1 (PMEPA1) is classified as a type 1β transmembrane protein with luminal, membrane spanning, and cytoplasmic domains [112]. Prostate transmembrane protein androgen induced 1 (PMEPA1), a TGF-β-responsive gene, inhibits TGF-β pathway via a negative feedback loop. Additionally, several studies have demonstrated that the PMEPA1 gene modulates other signaling cascades, including p53, EGF, Wnt, and Hippo signaling to interfere with tumorigenesis [113,114,115]. Additionally, galunisertib has been reported to be a promising drug under clinical investigation for treatment in HCC patients [116]. Therefore, understanding the secretion of TGF-β, which is defined as a mediator of the switch from a tumor-suppressive to a pro-oncogenic status, after galunisertib treatment is central to understanding the influence of galunisertib on tumorigenesis.

## 11. Discussion

The inflammatory microenvironment of liver tumors possesses a crucial role in facilitating HCC by inducing liver fibrosis, epithelial–mesenchymal transition, tumor invasion and metastasis [117]. Several types of cytokines and chemokines modulate the interaction between infiltrated immune cells and liver cancer cells, which in turn leads to remodeling of the liver microenvironment into profibrotic, proinflammatory, and proangiogenic signalings and thus becomes a tumor microenvironment [118].

Previously, Chhibar et al. reported that tumors are highly correlated with inflammatory liver diseases [119]. Several crucial inflammatory mediators, such as IL-6 and TNF-α, have been detected in the serum of HCC patients [120]. Higher levels of IL-6 in HCC patient serum are closely related to shorter survival times, suggesting the value of targeting inflammation-related molecules in HCC [120]. Moreover, increased levels of inflammatory cytokines and chemokines are observed in HCC patients with high levels of TNF-α [121,122]. Although TNF-α has the ability to induce tumor cell lysis, accumulating evidence shows that it plays critical roles in both tumor initiation and tumor development [122,123,124]. Additionally, dysregulation of TNF-α has been detected in many cancers, such as ovarian and renal cancers [122,123,124], as well as in the serum of patients with cancer, but not in that of healthy individuals [121,125].

Recently, the IL-6 concentration in serum has been defined as a promising tumor marker for HCC [126,127,128]. A high level of IL-10 in serum has been shown to be associated with poor survival in HCC patients undergoing surgical resection and in patients with unresectable tumors [129,130]. Similarly, the levels of IL-8 and IL-18 in serum have been shown to be useful markers of tumor invasiveness in HCC patients [131,132]. In addition, Jang and colleagues found that the levels of circulating inflammatory cytokines, such as TNF-α, interferon-c (IFN-c), IL-4, IL-6, and IL-10, are highly correlated with tumor stage, tumor response, and patient survival in HCC, and multivariate analysis showed that the IL-6 level is an independent indicator of unfavorable prognosis [120]. Furthermore, several reports have demonstrated that IL-6 is a cytokine with significant predictive ability for HCC patient survival and is associated with tumor size and aggressiveness [127,133]. IL-6 has also been shown to result in a highly metastatic potential in HCC and decrease apoptosis [127,133]. Additionally, the blood levels of prometastatic cytokines, such as TNF-α, IL-1, and IL-6, have been shown to be higher in HCC patients than in healthy individuals [19]. On the other hand, TGF-β1 expression has been found to be higher in malignant tumors, including HCC. TGF-β1 can cooperate with other cytokines, such as TNF-α, ILs, and IFNs, released from various liver cells and participates in various processes, including cell proliferation, apoptosis and inflammation [134].

In this review, we summarized data indicating that numerous chemotherapeutic drugs used in HCC patients induce the secretion of various cytokines, suggesting that inflammatory cytokines might play important roles in modulating drug resistance to HCC (Table 2). Moreover, the tumor microenvironment plays crucial roles in influencing cytokine secretion and drug resistance in HCC (Figure 1). Therefore, we need to investigate the relationship between chemotherapeutic agents and cytokine secretion and the mechanisms of secreted cytokines in more detail to counteract drug resistance in HCC. Currently, various drugs, such as axitinib, brivanib, bevacizumab, cetuximab, erlotinib, linifanib, and sunitinib, are in different phases of clinical trials [135]. Six systemic chemotherapeutic drugs have been approved according to phase III trials, which are expected to cure HCC patients at all stages via combination therapies of two immunotherapy regimens [136]. Hopefully, these drugs can one day be used in patients to counteract drug resistance and enhance therapeutic efficiency.

## Figures and Tables

**Figure 1 ijms-22-13627-f001:**
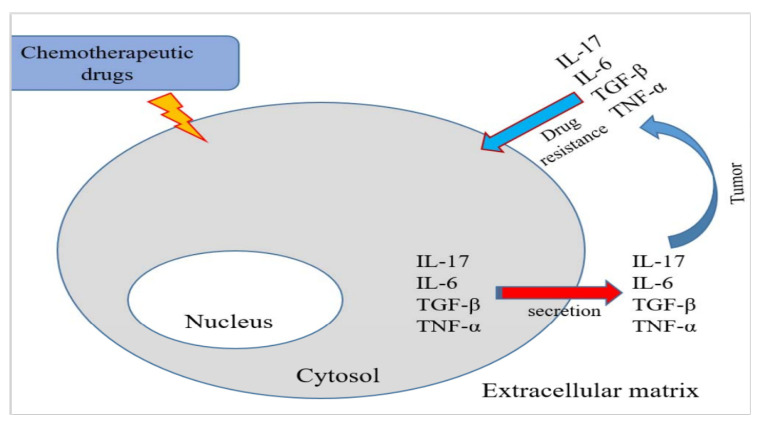
We summarized cytokine secretion in hepatocellular carcinoma (HCC) treated with various chemotherapeutic drugs. When cells were treated with chemotherapeutic drugs, several cytokines, including interleukin (IL)-17, IL-6, transforming growth factor-beta (TGF-β), and tumor necrosis factor-alpha (TNF-α), were secreted into the extracellular matrix. However, these secreted cytokines might modulate the tumor microenvironment directly or indirectly, in turn influencing the efficacy of chemotherapeutic drugs and finally resulting in drug resistance in HCC.

**Table 1 ijms-22-13627-t001:** The regulated mechanisms, pathways and effects of various chemotherapeutic drugs.

Drug	Mechanism				Pathway		Effect	
Fluorouracil								
	Fluorodeoxyuridine monophosphate			↑			Cell proliferation	↑
	CDC and ADCC effects			↑			Apoptosis	↑
	Epithelial-mesenchymal transition			↑			Cell migration	↑
	17β-Estradiol (E2)			↑	IL-6/STAT3 signaling	↓	Cell proliferation	↓
Cisplatin								
					ATM-NF-kB pathway	↑	DNA repair, cisplatin resistance	↑
					ATM-NF-kB-SOX2 pathway	↑	Stemness	↑
					STAT3 pathway	↓	Tumor growth	↓
	Cleaved PRAP-1			↑			Apoptosis	↑
					ATR, p53, p73 and MAPK pathways	↑	Apoptosis	↑
Oxaliplatin								
	IL-17 secretion			↑	NF-κB, MAPK and PI3K pathways	↑	Regulation of autophagy	
					p53-caspase 8-caspase 3 cascade	↑	Apoptosis	↑
	IL-6 secretion			↑	NF-κB, MAPK and p38 pathways	↑	Inflammation	↑
Celecoxib								
	E-cadherin			↑	COX-2-PGE2-Akt-ERK cascade	↓	Cell motility	↓
	Epithelial-mesenchymal transition			↑			Inflammation	↓
	COX-2 expression			↑			Inflammation	↑
Doxorubicin								
	lncRNA *H19*			↑			Cell survival and proliferation	↓
	Cytochrome p450-3A4 (CYP3A4) enzyme			↑			Doxorubicin toxicity	↓
					MEK/ERK cascade	↑	Apoptosis	↑
					MEK/ERK pathway	↑	Inflammation	↑
Sorafenib								
	CCL22 expression			↑	TNF-α-RIP1-NF-κB pathway	↑	Epithelial-mesenchymal transition	↑
	IL-6Rα induction			↓			Sorafenib resistance	↓
	IL-6 secretion			↑	DNMT1-OCT4 pathway	↑	Tumor recurrence	↑
Infliximab								
	IL-1β, IL-6, IL-17			↓			Apoptosis	↑
Galunisertib								
	E-cadherin	↑	SKIL, PMEPA1	↓			Invasiveness	↑

**Table 2 ijms-22-13627-t002:** The dysregulated cytokines by various chemotherapeutic drugs.

Agent	Secreted Cytokine
Oxaliplatin	IL-17
Celecoxib	IL-6
Doxorubicin	TGF-β, IL-6
Sunitinib	IL-6
Infliximab	TNF-α
Galunisertib	TGF-β

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
