# Peer review of "Chemotherapeutic Drug-Regulated Cytokines Might Influence Therapeutic Efficacy in HCC"

_ijms, 2021, doi:10.3390/ijms222413627_

Round 1
Reviewer 1 Report
Authors described that blocking TNF-α production could be a suitable approach to enhance the effect of classical chemotherapy in HCC patients. Authors should clearly describe the mechanism how TNF-αdecreases the effect of chemotherapy.
Authors should describe how IL-17/IL-17R binding inhibits oxaliplatin-induced cell death.
What is the mechanism how upregulated E-cadherin induced by COX-2 promotes apoptosis and inhibits cell proliferation.
Author Response
1.Authors described that blocking TNF-α production could be a suitable approach to enhance the effect of classical chemotherapy in HCC patients. Authors should clearly describe the mechanism how TNF-αdecreases the effect of chemotherapy.
Thanks for the great comment. (p2, lines 80-82)
To date, there have no detailed evidence indicate the mechanism how TNF-α decreases the effect of chemotherapy. We have found some information about the relationship between TNF-α and the effect of chemotherapy. Infliximab, an anti-TNF-α antibody, can increase Fluorouracil -induced the levels of cleaved caspase-3 in the presence of active complement in HCC [1]. Moreover, based on the clinical parameter correlation analyses, hepatitis B virus (HBV) or HCV infection and high survival rate are significantly related to TNF-α high expression (P-value=0.048 and 0.007, respectively) using chi-square analysis. Survival analysis showed that HCC patients with high TNF-α have shorter survival time than those low TNF-α expression (P=0.004), suggesting that TNF-α accelerates HCC development. Consistently, Cox regression analysis indicated that TNF-α is an independent predictor of poor survival of HCC patients (P<0.05, HR=1.689, 95%CI: 1.015–2.810) [1].
- Authors should describe how IL-17/IL-17R binding inhibits oxaliplatin-induced cell death.
We have added new information. (p4, lines 158-164)
Wu and colleagues have demonstrated that the increased Bcl-2 and decreased Bax are observed in HCC cells using Western blot after oxaliplatin treatment with IL-17 stimulation, However, the effect is abolished after stimulation with anti-IL-17 antibody. Moreover, IL-17 can induce the levels of p-JAK2 and p-STAT3 with oxaliplatin treatment. Based on the evidence, IL-17 interacts with IL-17R can decrease oxaliplatin-induced cell death through the JAK2-STAT3 cascade [2].
3.What is the mechanism how upregulated E-cadherin induced by COX-2 promotes apoptosis and inhibits cell proliferation.
Thanks for reviewer great comment.
The effect of the regulation between E-cadherin and COX-2 that we mentioned is described below. E-cadherin protein by via the inhibition of the Cyclooxygenase-2 (COX-2) prostaglandin E2 (PGE2)-p-Akt/p-ERK cascade (Table 2)[3]. COX-2, a rate-limiting enzyme, plays important roles in the process of inflammation-tumor transformation and the sequential oxygenation of arachidonic acid (AA) to synthesize prostaglandins and thromboxanes [4]. To date, there have no detailed evidence indicate that COX-2 can induce E-cadherin expression to promote apoptosis and inhibit cell proliferation. It needs more investigation and evidence to address the great issue.

Reviewer 2 Report
Given that it is 2021, the bibliography of the entire manuscript is not up to date. 29% of bibliography is from last 5 years (2016-2021).
HCC epidemiology source is from 2005, while the latest data are available [1-3]. This is unacceptable. Therefore I am not convinced that the remaining data described are up-to-date. I believe the work must be updated and the data must be re-examined. At the moment, I cannot agree to its publication.
[1] International Agency for Research on Cancer. GLOBOCAN 2018. IARC WHO
https://gco.iarc.fr/today/online-analysis-map?v=2020&mode=population&mode_population=continents&population=900&populations=900&key=asr&sex=0&cancer=11&type=0&statistic=5&prevalence=0&population_groupearth&color_palette=default&map_scale=quantile&map_nb_colors=5&continent=0&rotate=%255B10%252C0%255D (2020)
[2] Llovet JM, Kelley RK, Villanueva A et al. Hepatocellular carcinoma. Nat Rev Dis Primers 2021, 7(6). https://doi.org/10.1038/s41572-020-00240-3
[3] Konyn P, Ahmed A, Kim D. Current epidemiology in hepatocellular carcinoma. Expert Rev Gastroenterol Hepatol. 2021 Nov;15(11):1295-1307. doi: 10.1080/17474124.2021.1991792. Epub 2021 Oct 22. PMID: 34624198.
Author Response
Given that it is 2021, the bibliography of the entire manuscript is not up to date. 29% of bibliography is from last 5 years (2016-2021). HCC epidemiology source is from 2005, while the latest data are available [1-3]. This is unacceptable. Therefore I am not convinced that the remaining data described are up-to-date. I believe the work must be updated and the data must be re-examined. At the moment, I cannot agree to its publication.
[1] International Agency for Research on Cancer. GLOBOCAN 2018. IARC WHO
https://gco.iarc.fr/today/online-analysis-map?v=2020&mode=population&mode_population=continents&population=900&populations=900&key=asr&sex=0&cancer=11&type=0&statistic=5&prevalence=0&population_groupearth&color_palette=default&map_scale=quantile&map_nb_colors=5&continent=0&rotate=%255B10%252C0%255D (2020)
[2] Llovet JM, Kelley RK, Villanueva A et al. Hepatocellular carcinoma. Nat Rev Dis Primers 2021, 7(6). https://doi.org/10.1038/s41572-020-00240-3
[3] Konyn P, Ahmed A, Kim D. Current epidemiology in hepatocellular carcinoma. Expert Rev Gastroenterol Hepatol. 2021 Nov;15(11):1295-1307. doi: 10.1080/17474124.2021.1991792. Epub 2021 Oct 22. PMID: 34624198.
Thanks for reviewer comment. (p10, lines 411-413)
Thanks again for reviewer providing the literature, we have cited the reference in the manuscript. Because there have few literatures about the information between chemotherapy and inflammation/inflammatory cytokine. Hence, we only provide 29% of bibliography of the issue from last 5 years in this manuscript.

Round 2
Reviewer 1 Report
I do not have any more comments.
Reviewer 2 Report
Despite the assurances, the references have still not been corrected. Unfortunately, it is also not true that no newer publications on the subject have appeared. I am sorry, I cannot agree to publication.